# Role of Citrus Fruit Juices in Prevention of Kidney Stone Disease (KSD): A Narrative Review

**DOI:** 10.3390/nu13114117

**Published:** 2021-11-17

**Authors:** Yazeed Barghouthy, Bhaskar K. Somani

**Affiliations:** 1GRC n°20, Groupe de Recherche Clinique sur la Lithiase Urinaire, Hôpital Tenon, Sorbonne Université, F-75020 Paris, France; yazeedmail@gmail.com; 2Department of Urology, University Hospital Southampton NHS Trust Southampton, Southampton SO16 6YD, UK

**Keywords:** orange, lemon, lemonade, grapefruit, citrate, lemonade, nephrolithiasis, kidney calculi, risk

## Abstract

To explore the relationship between citrus fruit juices (oranges, grapefruits, and lemonades) and kidney stone disease (KSD). Methods: A systematic review was performed using the Medline, EMBASE, and Scopus databases, in concordance with the PRISMA checklist for all English, French, and Spanish language studies regarding the consumption of citrus fruit juices and the relationship to urinary stone disease. The main outcome of interest was the association of citrus fruit juices with KSD. Results: Thirteen articles met the criteria for inclusion in the final review. Three large epidemiological studies found that grapefruit juice was a risk factor for stone formation, while orange juice did not increase the risk for KSD. Ten small prospective clinical studies found that orange, grapefruit, and lemon juices all increased urinary citrate levels. Only orange and grapefruit juices had an alkalinizing effect and while lemon juice has a protective effect by raising urinary citrate levels, it lacked a significant alkalinizing effect on urine pH. Orange juice and grapefruit juices significantly increased urinary oxalate levels, while orange juice also had a high carbohydrate content. Conclusion: While orange juice seems to play a protective role against stone formation, grapefruit was found to raise the risk of KSD in epidemiological studies but had a protective role in smaller clinical studies. Lemon juice had a smaller protective role than orange juice. Larger amounts of, as well as more accurate, data is needed before recommendations can be made and a high carbohydrate content in these juices needs to be taken into consideration.

## 1. Introduction

Kidney stone disease (KSD) has increased in prevalence in recent decades, affecting around 10–14% of the population in industrialized nations. This increase is in part attributed to the dietary changes that occurred in the last century worldwide [1,2]. In addition, the distribution of stone types has also been affected by diet. While calcium oxalate still comprises the majority of stones, the percentage of uric acid stones has risen, mainly due to the high sugar content of the modern diet and increasing numbers of patients with metabolic syndrome (MetS) [3,4].

The EAU and AUA guidelines provide dietary recommendations to avoid stone formation and recurrence. Accordingly, moderation of salt, protein, oxalate, uric acid, and calcium consumption is needed, along with the necessary fluid intake to produce at least 2–2.5 L of urine per day [5,6].

Citrus fruits, including oranges, grapefruits, and lemons, and the juices produced from these fruits, are central components of the modern diet and enjoy growing popularity with recent health trends. The consumption of these juices might influence the evolution of kidney stones in several aspects. On the one hand, the protective effects include the liquid intake itself entailed with juice consumption, the high citrate content, and the alkalinizing effect on urinary pH. On the other hand, the high carbohydrate or sugar content in these juices might raise that risk for KSD [7,8,9].

In this paper, we will review the available literature, evaluating the relationship between citrus fruit juices consumption and KSD. In the discussion, we will explore the different components of citrus fruits and their lithogenic or protective effect against urinary stone formation.

## 2. Methods

### 2.1. Evidence Acquisition

#### 2.1.1. Inclusion Criteria

Studies in English and French exploring the relation between orange, grapefruit, and lemon juice consumption with kidney stone disease (KSD).

Studies in adults published up to January 2021.

#### 2.1.2. Exclusion Criteria

Meta-analyses, systematic reviews, case reports, editorials and letters, comments, and abstracts.

### 2.2. Search Strategy and Study Selection

A systematic review was performed, using the PUBMED-MEDLINE, EMBASE, and Scopus databases. The Preferred Reporting Items for Systematic Reviews and Meta-Analyses (PRISMA) statement (Figure 1) was followed in this review. The review was not prospectively registered. The keywords used were the following: “orange”, “lemon”, “lemon water”, “grapefruit”, “citrate”, “citric”, “lemonade”, “nephrolithiasis”, “urinary stones”, “kidney stones”, “urolithiasis”, and “risk”. Search terms included a combination of keywords above with each of the following: urinary stones, nephrolithiasis risk (for example, “orange AND nephrolithiasis”). No time period restriction was set. Two reviewers (YB, BS) identified all of the studies independently and any discrepancies were resolved with mutual consensus. A narrative synthesis, rather than a quantified meta-analysis of data, was performed due to the heterogeneity of outcomes.

## 3. Results

Of 1248 articles found through the initial database search, 127 abstracts were reviewed, of which, 46 full text articles were assessed for eligibility and 13 articles met the criteria for inclusion in the final review (Figure 1). Table 1 summarizes the most relevant studies found on this topic. Large epidemiological studies found that grapefruit was a risk factor for stone formation, while orange juice did not raise the risk for KSD [10,11,12].

More specifically, Curhan et al. found that the risk of stone formation in a large male population increased by 37% for each 240-mL portion of grapefruit juice consumed daily. While the results showed a more protective role for orange juice, it was not statistically significant [10]. In the Nurses’ Health study, risk of stone formation with grapefruit juice consumption was 44% [11]. In a third study comprising three large cohorts, Ferraro et al. showed that orange juice was protective against stone formation (12% risk reduction; 95% CI = 2% to 21%), but the previously demonstrated risk for grapefruit was not shown in this study [12].

Small prospective clinical studies found that orange, melon, grapefruit, and lemon juices increased urinary citrate levels [13,14,15,16,17,18,19,20,21]. Baia et al., for example, showed that citrate increased after 4 h of ingestion to 0.35 ± 0.15 vs. baseline of 0.22 ± 0.10 mg/mg creatinine in the melon group, 0.32 ± 0.17 vs. baseline of 0.14 ± 0.09 mg/mg creatinine in the orange group, and to 0.29 ± 0.21 vs. baseline of 0.15 ± 0.12 mg/mg creatinine in the lime group. However, only orange, melon, and grapefruit juices had significant alkalinizing effects [13,15].

Orange juice increased urinary oxalate and did not alter calcium excretion [17]. Grapefruit juice also significantly increased urinary oxalate levels [19]. Lemonade did not increase urinary citrate nor pH levels in one study [22]. Table 2 summarizes the changes in urinary parameters after intake of the various juices in the reviewed studies.

## 4. Discussion

Prevention of recurrent kidney stone disease requires both diet and lifestyle modification, along with medical management in some patients [23]. The cornerstones for prevention are adequate hydration, prevention of urinary oxalate and uric acid overload, regulation of urinary pH to prevent acidic urine, and avoiding low urinary citrate levels [24].

The consumption of citrus fruit juices became increasingly popular in the last decades because of their branding as healthy alternatives to other carbohydrate-rich soft drinks, with the recommended quantity of consumption for orange juice at 200 mL daily. In addition, their consumption was also promoted as an alternative to pharmacological therapy with potassium citrate, which presents compliance challenges for many patients due to the treatment regimens [13].

In this review, three large epidemiological studies [10,11,12] showed an increased risk for urinary stone formation with the ingestion of grapefruit juice and a protective role for the consumption of orange juice. Lemonade consumption was not evaluated in these large cohort studies. Smaller prospective clinical studies did not demonstrate the increased risk for grapefruit juice [14,19,20] and showed the protective effects on urinary parameters for orange juice [13,14,15,17,18], while the results for lemon juice were mixed [13,16,21,22].

### 4.1. Role of Citrus Juices, Mechanism of Action, and Comparison between Studies

For clarification purposes, the terms lemon juice and lemonade may be confusing and were variable in different studies. In one study, lemonade was prepared with reconstituted lemon juice mixed with tap water to a total volume of 2:1 [16]. In three studies, fresh lemon juice was mixed with water and a sweetener [13,21,22]. In two other studies, lemonade was ready-made and contained lemon juice concentrate in addition to water and fructose corn syrup [15,18].

Citrus fruit juices exert their potential protective effect in multiple ways. The first and most obvious one is the beneficial addition of fluid intake itself with citrus juice consumption [25,26]. The second is the citrate content in citrus fruit juices, which is one of the strongest inhibitors of urinary stone formation [27,28]. Citric acid, derived from citrus juices, is a tricarboxylic acid which exists mainly in the form of the salt citrate, at physiologic blood pH (7.4) [29]. Citrate is both an endogenous byproduct of the citric acid (Krebs) cycle and a nutrient absorbed in the diet. Citrate is metabolized in the liver to bicarbonate, comprising an alkali load that reduces renal reabsorption of citrate, thus allowing more citrate to be later excreted in the urine [30,31].

In the urine, citrate inhibits the spontaneous nucleation and agglomeration of calcium oxalate crystals [14,15]. This action of citrate in the urine is particularly important, given the fact that many KSD patients suffer from hypocitraturia [28,29]. Important causes for this include renal tubular acidosis, urinary tract infections, and bowel disease, contributing to a metabolic acidosis thereby consuming the body citrate storage, in addition to the use of certain medication, such as acetazolamide and thiazide [8,29,32].

In certain urinary stones, high urine calcium levels constitute the main risk factor for stone formation. These include mainly calcium oxalate dehydrated stones-Class II, according to the Daudon’s classification [33] and also calcium phosphate stones. Citrate plays a crucial role in lowering urine calcium levels by sequestration of Ca^+2^ ions through the formation of citrate–calcium complexes, which are more soluble in the urine at physiological urine pH levels [34].

Although all citrus fruits are a particularly rich source of citrate, the demonstrated difference in their protective effect against urinary stone formation might be the result of the different content of citrate in each fruit [35,36]. Moreover, the citrate content in these fruits does not automatically translate into the expected rise in urinary citrate. For example, Odvina demonstrated that although orange and lemon juices had comparable citrate content, orange juice consumption resulted in a significantly higher urinary citrate excretion when compared to lemonade [15].

Another factor for the protective effects of citrus juices is the alkalinizing effect on urine pH, which plays a particularly important role against calcium oxalate and uric acid stone formation [13]. Citrus juices are acidic by definition; however, they have an acidifying effect on the organism and an alkalinizing effect on the urine [16,28,30]. This alkalinizing effect is thought to originate, in part, by the metabolism of citrate in the body, with the production of bicarbonate molecules in this process and filtration of this bicarbonate load in the urine.

Uric acid stones, for example, are principally influenced by low diuresis, hyperuricosuria, and last but certainly not least, low urine pH, which are situations that contribute to systemic acidosis. These situations are variable and include high consumption of animal protein, chronic diarrhea, or laxative abuse, that contribute to gastrointestinal alkali loss and metabolic syndrome including obesity and diabetes mellitus [37]. Acidic urine pH, in turn, favors the formation of the relatively insoluble urate crystals in urine [33]. The alkalinizing effects of citrate on the urine thus plays a fundamental role in protection against uric acid stone formation. Citrate is even used in medical dissolution therapy for uric acid stones, allowing certain patients complete resolution of their stones and avoidance of surgical procedures to remove the stones [38].

Another mechanism for urine alkalinization is the concomitant intake of potassium with consumption of citrus fruit juices. This potassium load increases the tubular secretion of potassium (K^+^) ions in exchange for sodium (Na^+^) ions, instead of proton (H^+^) ion secretion. Together with the secreted bicarbonate, these two mechanisms contribute to urine alkalinization [39].

This alkalinizing effect, however, is variable between different citrus juices, mainly due to the difference in citrate and potassium levels, in addition to the importance of the type of accompanying cations and their content. Orange juice, for example, had the highest alkalinizing effect compared to lemon and the ingestion of lemon juice did not change the pH of urine significantly [15]. Other studies showed that this is also caused, in part, by the fact that orange citrate complexes with potassium ions, while citrate from lemon complexes with proton (H^+^) ions, thus neutralizing the alkali load resulting from citrate ingestion and conversion to bicarbonate in the liver [13,18]. In addition, orange juice leads to lower ammonium secretion in the urine than lemon juice, leading to a difference in net acid secretion [15]. Care must be taken, as alkalinizing effect carries a potential theoretic risk of promoting the formation of calcium phosphate stones, but this is seen only in rare cases of high consumption of citrus juice.

Finally, there is a wide consensus in the urological guidelines that a high intake of dietary fiber from fruits and vegetables is associated with a reduced risk of kidney stones [40]. This was, in addition, recently demonstrated in a population-based prospective cohort study in the UK, in which fiber intake was among the protective factors against stone formation [41]. Fiber content in the citrus fruits potentially also contributes to their protective effect.

With all of these positive effects on urinary stone risk parameters, one would expect a clear protective role for citrus fruits against stone formation, although this does not always happen. The possible explanations are that not all citrus juices have the same urine alkalinizing effect, as mentioned above [15,18]. Secondly, citrus fruits carry variable potassium, calcium, and oxalate loads too and this might offset the positive effects of increasing urinary citrate and urine alkalinization [16]. Wabner and Pak showed that orange juice increased urinary oxalate and did not decrease calcium excretion. For comparison, they showed that potassium citrate decreased urinary calcium without altering urinary oxalate. Thus, orange juice did not have the same capacity as potassium citrate to decrease urinary saturation of calcium oxalate [17]. Grapefruit juice also significantly increased urinary oxalate levels. However, this was not associated with an increased risk of supra-saturation of calcium oxalate, calcium phosphate, or uric acid, probably due to the balancing effect of the increase in urinary citrate levels [19]. Hönow et al. evaluated grapefruit juice and compared its effects on urinary parameters in comparison to orange juice and concluded that there is no component of grapefruit juice responsible for an increasing risk of stone formation [14]. In this study, the BONN risk index (BRI) was used to evaluate the crystallization risk after intake of grapefruit, orange, and apple juices. The BRI was found to be significantly decreased after the intake of 1 L of juice. This was mainly attributed to the increase in urine citrate and urine pH. The absence of increased crystallization risk for CaOx was also echoed in other studies as well [15,17,19,20].

An important factor, largely underestimated, is the high sugar content of citrus fruit juices. Orange juice carries a high load of glucose and fructose, estimated between 21 g (in lemonade) to 26 g (in orange juice) of carbohydrates per cup [9]. This high sugar content can lead to hypercalciuria, in addition to promoting insulin resistance in the long term [42,43]. This is also important when evaluating lemonade, given that in many studies, lemonade was prepared with added sugar or artificial sweeteners [13,18,21,22].

The above factors combined contribute to understanding the effect citrus fruit juices have on urinary factors. They also highlight the reasons for the mixed results between different studies regarding the protective role of different citrus fruits against stone formation.

### 4.2. Strengths, Limitations and Areas of Future Research

Our systematic review assesses the role of citrus fruits on developing KSD and is done according to the PRISMA checklist, summarizing the data on this topic. It also gives an overview into the results and pathophysiology associated with it. However, the heterogeneity of included studies did not allow formal meta-analysis. On one hand, large epidemiological studies were affected by selection and recall biases. On the contrary, prospective clinical studies included short term intervention in only a small number of healthy volunteers or stone forming patients, without a control group. These different populations (stone formers versus healthy volunteers) probably also have different baseline characteristics in their urinary risk factors and metabolism of citrate. In addition, the reviewed studies lack a uniformity of consumption regimens which might explain the different results between small prospective studies and large cohort studies. Larger studies with well-defined protocols, matched controls, and clear regimens of citrus juice consumption and longer follow-up, are needed before any definitive conclusions can be made.

## 5. Conclusions

Orange juice seems to play a protective role against stone formation, but the high sugar content needs to be taken into consideration. Grapefruit was found to raise the risk of KSD in epidemiological studies; however, smaller clinical studies found a protective role against stone formation. Lemon juice had a protective effect by raising urinary citrate levels but lacked a significant alkalinizing effect on urine pH. While citrus fruit juices were generally found to be protective, larger studies with more clinical data are needed for a better understanding of citrus fruit juices on urinary risk factors of stone formation.

## Figures and Tables

**Figure 1 nutrients-13-04117-f001:**
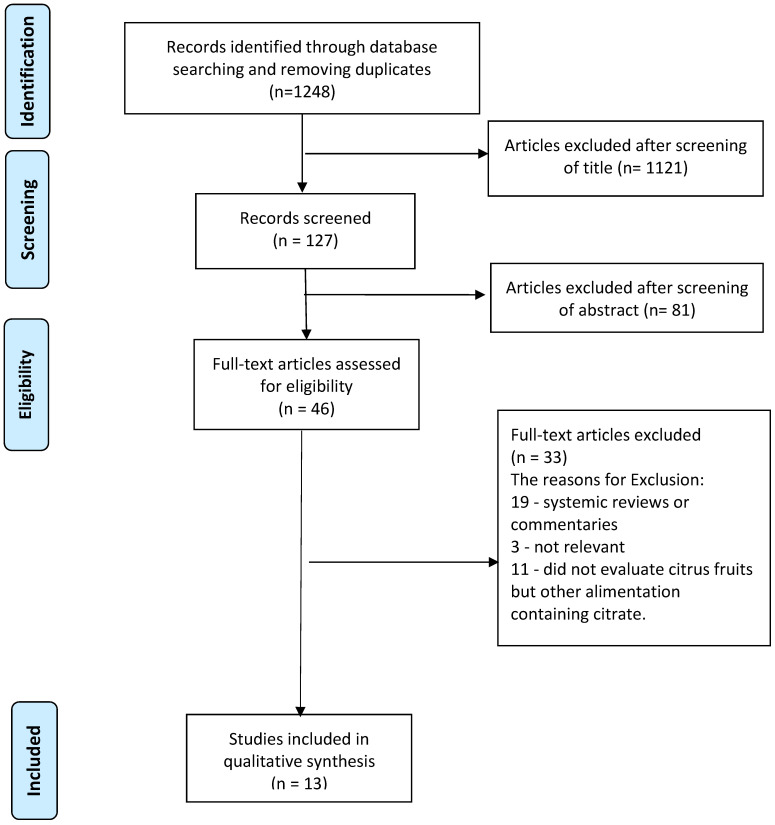
PRISMA flowchart of the included studies—please see revised PRISMA chart in a separate file attached.

**Table 1 nutrients-13-04117-t001:** Summary of the main experimental studies of orange, grapefruit, and lemon juice intake and urinary stone formation.

Author.	Year	Study Type	Orange/Grapefruit/Lemon	Sample Size	Assessment	Study Goal	Conclusion
Curhan [10]	1996	Prospective cohort study	Orange, Grapefruit	45,289	Questionnaire	The relation between intake of 21 different beverages and the risk of symptomatic kidney stones in men.	Grapefruit juice was directly associated with risk of stone formation. No increased risk for orange juice consumption.
Curhan [11]	1998	Prospective cohort study	Orange, Grapefruit	81,093	Questionnaire	The association between the intake of 17 beverages and risk for kidney stones in women.	Grapefruit juice was directly associated with risk of stone formation. No increased risk for orange juice consumption.
Ferraro [12]	2013	Prospective cohort study	Orange, Grapefruit	217,883	Questionnaire	The association between intake of several types of beverages and incidence of kidney stones.	Protective effect for the consumption of orange juice. Risk of grapefruit juice not significant.
Baia [13]	2012	Prospective clinical study	Orange, Lime	30 hypocitraturic stone forming patients	Urine analyses	To compare the acute effects of a non-citrus fruit (melon) vs. citrus fruits (orange and lime) on citraturia and other lithogenic parameters.	Significant and comparable increases of mean urinary citrate were observed in all groups. Mean urinary pH significantly increased after orange juice but not after lime juice consumption.
Hönow [14]	2003	Prospective clinical study	Orange, Grapefruit	9 non-stone formers	Urine analyses	To evaluate the influence of grapefruit and apple juice consumption on urinary variables and crystallization, in comparison with orange juice.	Both grapefruit juice and apple juice reduce the risk of CaOx stone formation at a magnitude comparable with the effects obtained from orange juice.
Odvina [15]	2006	Prospective randomized study	Orange, Lemonade	13 volunteers (9 healthy and 4 stone formers)	Blood and urine analyses	To compare the effects of orange juice with those of lemonade on acid–base profile and urinary stone risks under controlled metabolic conditions.	Orange juice has greater alkalinizing and citraturic effects than lemonade. Orange juice is associated with lower calcium-oxalate supersaturation and lower uric acid.
Seltzer [16]	1996	Prospective clinical study	Lemonade	12 hypocitraturic stone formers	Urine analyses	To evaluate the urinary biochemical effects of dietary citrate supplementation (lemonade).	Lemonade results in elevated urinary citrate levels and can be a possible treatment in hypocitraturic calcium stone formers.
Wabner [17]	1993	Prospective clinical study	Orange	13 volunteers (8 healthy and 3 hypocitraturic stone formers)	Urine analyses	To evaluate the urinary biochemical effects of orange juice compared to potassium citrate.	Compared to potassium citrate, orange juice caused a similar increase in urinary pH and urinary citrate but increased urinary oxalate and did not reduce calcium excretion.
Large [18]	2020	Prospective randomized study	Orange, Artificial lemonade	10 non-stone formers	Urine analyses	To evaluate urinary citrate and pH changes with consumption of low-calorie orange juice and artificial lemonade.	Daily consumption of orange juice can raise urinary pH.
Goldfarb [19]	2001	Prospective clinical study	Grapefruit	10 non-stone formers	Urine analyses	To study the basis of the lithogenic effect of grapefruit juice demonstrated in epidemiologic studies.	Grapefruit juice associated with an increase in mean oxalate and citrate excretion. However, no net change in calculated supersaturation or lithogenicity.
Trinchieri [20]	2002	Prospective clinical study	Grapefruit	7 non-stone formers	Urine analyses	To investigate changes in urinary stone risk factors after consumption of grapefruit juice.	Grapefruit juice significantly increases urinary excretion of citrate, calcium, and magnesium.
Penniston [21]	2007	Retrospective analysis	Lemonade	100 CaOx stone formers	Urine analyses	To evaluate the urinary biochemical effects of lemonade compared to K-citrate + lemonade.	Lemonade raised urinary citrate and total urine volume but was less effective than K-citrate + lemonade.
Koff [22]	2007	Prospective clinical trial	Lemonade	21 stone formers	Urine analyses	To evaluate the urinary biochemical effects of lemonade compared to potassium citrate.	Lemonade did not increase urinary citrate or pH levels.

**Table 2 nutrients-13-04117-t002:** Summary of the important urinary parameter changes in the reviewed articles (U—urine, N/A—not available). Changes are significant unless otherwise stated with NS (Non Significant).

	U-pH	U-Citrate	U-Potassium	U-Oxalate	U-Calcium	CaOx Crystallization Risk
Baia [13] - Melon - Orange - Lime	Increase Increase NS change	Increase Increase Increase	NS Increase Increase NS change	Not measured (N/A)	N/A	N/A
Hönow [14] - Grapefruit - Orange - Apple	NS Increase Increase Increase	Increase Increase Increase	Increase Increase Increase	NS Increase NS Increase NS Increase	Decrease Decrease NS decrease	Decrease NS Decrease NS Decrease
Odvina [15] - Lemonade - Orange	NS change Increase	NS Increase Increase	NS change Increase	No Significant Change Increase	NS Decrease NS Decrease	NS Decrease Decrease
Seltzer [16] - Lemonade	Not Measured	Increase	N/A	NS change	NS Decrease	N/A
Wabner [17] - Orange	Increase	Increase	Increase	Increase	NS Decrease	NS Change
Large [18] - Commercial Orange and Lemonade Juices	Increase	Increase	NS Increase for Orange Juice No Change for Lemonade	NS Increase for Orange Juice No Change for Lemonade	NS changes	N/A
Goldfarb [19] - Grapefruit	NS Increase	Increase	Increase	Increase	NS Decrease	No Change
Trinchieri [20] - Grapefruit	NS Decrease	Increase	NS Decrease	NS Increase	NS Increase	No Change

## Data Availability

Data is available and can be released on request.

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
