# Peer review of "Role of Citrus Fruit Juices in Prevention of Kidney Stone Disease (KSD): A Narrative Review"

_nutrients, 2021, doi:10.3390/nu13114117_

Round 1

Reviewer 1 Report

I do not see any need for corrections, exept in "References " section where position [13] merged with [12} and [15] with [14], but it is of course minor editorial error.

Pity that there are so few studies involving larger populations, both people with urolithiasis and controls.

Author Response

Comments and Suggestions for Authors

I do not see any need for corrections, except in "References " section where position [13] merged with [12} and [15] with [14], but it is of course minor editorial error.

Answer: We have now corrected this as per the reviewer’s recommendation.

Pity that there are so few studies involving larger populations, both people with urolithiasis and controls.

Answer: We agree that some larger high quality studies are needed in this area.

Reviewer 2 Report

First of all, this manuscript is rather a literature review but NOT a systematic review. There is neither predefined PICO/PECO nor quantitative or qualitative assessment of the literature. Results are just the list of the literature and discussion is just descriptive but not analytical.

Author Response

Comments and Suggestions for Authors

First of all, this manuscript is rather a literature review but NOT a systematic review. There is neither predefined PICO/PECO nor quantitative or qualitative assessment of the literature. Results are just the list of the literature and discussion is just descriptive but not analytical. 

Answer: We respectfully disagree with the reviewer. Although the review was not registered, this has been done in a systematic review methodology taking into account all aspects of PRISMA framework. The results are heterogenous and hence a meaningful meta-analysis was not possible. If the reviewer or editorial team strongly feel that we need to change the title, we will be happy to call it a narrative review.

Reviewer 3 Report

This systematic review is a very solid overview of an interesting topic which frequently comes up in clinical practice. The authors assessed the extent to which citrus fruit juices alter stone risk and urinary parameters. The presentation of the methods and results is succinct, and the discussion is substantive.

I recommend a few review protocol and results clarifications:

  • Was the review protocol prospectively registered (e.g., through PROSPERO)? If so, protocol number should be listed. If not, it should be stated that the review was not prospectively registered.
  • In Figure 1, please indicate the reasons for excluding articles (e.g., number of articles excluded because they were relevant to this review, number of articles excluded because they were case reports, etc.)
  • In Table 2 or in Methods, please provide definitions of the descriptors of urinary parameter change (e.g., increased, increased significantly, non-significant decrease, etc.). Please clarify whether you are referring to statistically significant increases/decreases. In either Table 2, the Results, or the Discussion, it would be worthwhile to give examples of these changes (e.g., “urinary citrate increased by 50 mg/day”).

Author Response

Comments and Suggestions for Authors

This systematic review is a very solid overview of an interesting topic which frequently comes up in clinical practice. The authors assessed the extent to which citrus fruit juices alter stone risk and urinary parameters. The presentation of the methods and results is succinct, and the discussion is substantive.

Answer: We thank the reviewer their very positive and supportive comments to our paper.

I recommend a few review protocol and results clarifications:

  • Was the review protocol prospectively registered (e.g., through PROSPERO)? If so, protocol number should be listed. If not, it should be stated that the review was not prospectively registered.

Answer: We have now done this in the manuscript, in the methods section, which reads –

The review was not prospectively registered.

  • In Figure 1, please indicate the reasons for excluding articles (e.g., number of articles excluded because they were relevant to this review, number of articles excluded because they were case reports, etc.)

Answer: We have now done this as per the reviewer’s recommendation.

  • In Table 2 or in Methods, please provide definitions of the descriptors of urinary parameter change (e.g., increased, increased significantly, non-significant decrease, etc.). Please clarify whether you are referring to statistically significant increases/decreases. In either Table 2, the Results, or the Discussion, it would be worthwhile to give examples of these changes (e.g., “urinary citrate increased by 50 mg/day”).

Answer: We thank the reviewer for their comment. We have now modified the paper accordingly.  In the results section, we added a paragraph as an example for the citrate increases according to Baia’s [13] results, however, due to the length of such examples, we refrained from giving more detailed numerical data for each parameter.

‘Baia et al, for example, showed that citrate increased after 4 hours of ingestion to 0.35 ± 0.15 vs. baseline of 0.22±0.10 mg/mg creatinine in the melon group, 0.32±0.17 vs. baseline of 0.14±0,09 mg/mg creatinine in the orange group, and to 0,29±0.21 vs. baseline of 0.15±0.12 mg/mg creatinine in the Lime group.     However, only orange, melon and grapefruit juices had a significant alkalinizing effect [13,15].’

In certain urinary stones, high urine calcium levels constitute the main risk factor for stone formation. These include mainly in calcium oxalate dehydrate stones -Class II according to the Daudon’s classification- [33], and also calcium phosphate stones. Citrate plays a crucial role in lowering urine calcium levels by sequestration of Ca+2 ions through the formation of citrate-calcium complexes, which are more soluble in the urine at physiologic urine pH levels [34].

Uric acid stones for example, are principally influenced by low diuresis, hyperuricosuria, and last but certainly not least, low urine pH and situation that contribute to systemic acidosis. These situations are variable and include high consumption of animal protein, chronic diarrhea or laxative abuse that contribute to gastrointestinal alkali loss, and metabolic syndrome including obesity and Diabetes mellitus [37]. Acidic urine pH in turn favors the formation of the relatively insoluble urate crystals in urine [33]. The alkalinizing effects of citrate on the urine thus plays a fundamental role in protection against uric acid stones formation. Citrate is even used in medical dissolution therapy for uric acid stones, allowing certain patients complete resolution of their stones and avoidance of surgical procedures to remove the stones [38].

Round 2

Reviewer 2 Report

No further comment.